# Comparison of Artificial Intelligence Control Strategies for a Peristaltically Pumped Low-Pressure Driven Membrane Process

**DOI:** 10.3390/membranes12090883

**Published:** 2022-09-13

**Authors:** José-Luis Díez, Vicente Masip-Moret, Asunción Santafé-Moros, José M. Gozálvez-Zafrilla

**Affiliations:** 1Instituto Universitario de Automática e Informática Industrial, Universitat Politècnica de València, 46022 València, Spain; 2Centro de Investigación Biomédica en Red de Diabetes y Enfermedades Metabólicas Asociadas (CIBERDEM), Instituto de Salud Carlos III, 28029 Madrid, Spain; 3Barcelona Supercomputing Center, 08034 Barcelona, Spain; 4Institute for Industrial, Radiophysical and Environmental Safety (ISIRYM), Universitat Politècnica de València, 46022 València, Spain

**Keywords:** low-pressure driven process, peristaltic pump, microfiltration, intelligent control, artificial intelligence, modelling, fouling, humic acid

## Abstract

Peristaltic pumping is used in membrane applications where high and sterile sealing is required. However, control is difficult due to the pulsating pump characteristics and the time-varying properties of the system. In this work, three artificial intelligence control strategies (artificial neural networks (ANN), fuzzy logic expert systems, and fuzzy-integrated local models) were used to regulate transmembrane pressure and crossflow velocity in a microfiltration system under high fouling conditions. A pilot plant was used to obtain the necessary data to identify the AI models and to test the controllers. Humic acid was employed as a foulant, and cleaning-in-place with NaOH was used to restore the membrane state. Several starting operating points were studied and setpoint changes were performed to study the plant dynamics under different control strategies. The results showed that the control approaches were able to control the membrane system, but significant differences in the dynamics were observed. The ANN control was able to achieve the specifications but showed poor dynamics. Expert control was fast but showed problems in different working areas. Local models required less data than ANN, achieving high accuracy and robustness. Therefore, the technique to be used will depend on the available information and the application dynamics requirements.

## 1. Introduction

Peristaltic pumping is used in membrane processes, such as microfiltration (MF) or ultrafiltration (UF), when high sealing without contact of the solutions with the lubricating fluids or friction elements is required. Typical applications of this configuration are in food processing industry separations, membrane bioreactors, membrane medical applications, and laboratory experimentation.

The use of peristaltic pumps in membrane bioreactor systems also has the advantage of diminishing the effect of tearing stress. For example, anaerobic sludge membrane reactors using peristaltic pumping for pressure application and recirculation have been used to treat municipal or industrial waters [1,2]. This kind of pumping allows special membrane applications such as the use of fluidized glass beads to improve membrane performance [3].

In medical systems using membranes, mainly in artificial kidneys or microdialysis sensors, pumps must not introduce harmful agents and must maintain sterility. Peristaltic pumps can meet this requirement with suitable flow rate characteristics [4]. However, problems related to the formation of particles may occur. These particles may be caused by wear of the tube due to spallation [5] or degradation of the solution. An example of the latter situation is the formation of protein particles caused by the effect of the tube tearing on proteins adsorbed on the tube walls [6].

The combination of peristaltic pumps and membranes is also widely used on a laboratory scale. For example, multi-channel peristaltic pumps can perform many membrane experiments at the same time [7] or experiments in which two peristaltic pumps work in opposite directions [8].

### 1.1. Modelling of Peristaltically Pumped Low-Pressure Driven Membrane Systems

The development of control systems to meet permeate flow specifications is essential for the proper functioning of the above-mentioned applications.

Most control systems are, in some way, model based. Therefore, the existence of a computational model deduced from the physical behavior of the system can facilitate control development. However, modeling and control of peristaltically pumped MF or UF membrane systems can be more challenging than other membrane configurations.

The behavior and performance of peristaltic pumps can be approximately modeled by lumped models based on physical considerations [9] or by using computational fluid dynamics [10]. Disturbance models can also be used for control purposes [11]. However, let us suppose that more accurate time-varying models are needed for control. In that case, the models could be improved by taking into account changes in feed properties and mechanical characteristics of the tubing material.

Membrane modelling must describe the permeate flux and component rejection and its evolution over time. In MF, retention depends on particle size and pore size distribution. The flow through the pores can be described by Poiseuille’s law. In UF, modelling of solute and solvent transport is based on hydrodynamic equations describing hindered diffusion and convection [12]. For most processes, fouling adds additional resistance to flow. Moreover, the effect of fouling involves longer time dynamics than that those caused by feed or operational variations. To describe the different fouling mechanisms (pore blocking or gel layer formation), empirical models such as those developed by Hermia can be used [13]. Other modelling difficulties are the different fouling potential of the solutions [14], the important effect of spacer design on fouling [15], and the fact that the membrane performance does not fully recover after the cleaning procedures. For both processes (MF and UF), the situation can be more complex in the case of a non-constant flow [16], as in the case of the pulsating flow produced by a peristaltic pump.

### 1.2. Intelligent Control Approaches for Low-Pressure Membrane Systems

In conclusion, accurate modeling of peristaltically pumped low-pressure driven membrane systems is challenging due to the changes in system performance over time. In general, models derived from first principles with sufficient accuracy are very complex. Therefore, classical control techniques cannot be applied directly. On the other hand, those models simple enough to apply well-known control techniques cannot describe the long-term behavior of the system. This fact led the authors to consider the use of artificial intelligence (AI) and system identification methods as the most appropriate approach.

Most attempts to control peristaltically pumped membrane systems have come from the field of hemodialysis, where accurate control of fluid delivery is especially critical [17,18]. In this application, the use of hierarchical adaptive and supervisory control has allowed adjusting the pump inputs to patient monitoring data [19].

In other fields where MF or UF have been applied, the modeling and control focus on the interaction of the membrane and the pump, but on the membrane performance. Niu et al. have recently carried out a critical review on the use of different AI methods in fouling prediction [20]. They found that the most modeled features were transmembrane pressure, flux dynamics, and flux decline by fouling. They divided the AI techniques into single and hybrid algorithms. They indicated that the most employed single algorithm was artificial neural networks (ANN), but others such as fuzzy logic, genetic programming, or support vector machines were also used. Hybrid algorithms were usually built by combining these techniques with a search algorithm. Jawad et al. conducted a review that found that permeate flux is the most modeled feature in the different MF or UF applications. The same study showed that the most used model input is transmembrane pressure (TMP), and that the composition-related parameters were also used as input in the different works [21]. Recently, machine learning has been used to model the dynamics of filtration and backwashing of UF by a back propagation ANN [22]. Ultrafiltration of protein solutions has been effectively modeled using ANN for a tubular crossflow membrane [23]. This work used as inputs: operational time, pH, and ionic strength; and as outputs: filtrate flow and protein transmission. The authors compared the performance of ANN with that of the Hermia models [24] and obtained similar performance results for the fitted experiments, but better ANN extrapolation capability for experiments not included in the fitting process. Other AI-based modelling tools, such as fuzzy logic, have proven to be an alternative to ANN modelling [25].

### 1.3. Selection of Study Case for Comparison of Control Approaches

Given the promising results found in the literature on AI methods applied to low-pressure driven processes, this work aimed to compare the performance of a set of AI modeling techniques for controlling peristaltically pumped membrane processes subjected to strong fouling. The specific process case studied used a ceramic MF membrane with a pore size close to the maximum UF pore size range. The operating conditions of the MF system to be controlled were average crossflow velocity and average transmembrane pressure. Forced membrane fouling was expected to produce permeability variability due to both reversible and irreversible fouling.

The chosen techniques cannot only provide an accurate system description but can also deliver models that are good for subsequent control design under all the expected operating conditions. This balance between modeling and control could lead to a better performance of the controlled MF system. Therefore, a comparison between the available methods was made to determine which can be more effective in controlling the operating conditions in a changing membrane system.

In order to obtain the data necessary for the creation of the models, each experiment consisted of an operational step with fouling followed by a cleaning step. For the first step, the selected foulant was humic acid at a relatively high concentration. Humic acid fouling is one of the main factors limiting MF in water treatment [26,27] with humic acid aggregates being responsible for most of the fouling [28]. For the second step, it was considered that fouling by organic matter can be removed from ceramic membranes by cleaning-in-place (CIP) procedures using alkalis. When NaOH is used, typical concentrations are in the range of 0.5–2% wt. [29,30]. The combination of rapid fouling with humic acid and subsequent cleaning with NaOH allowed short dynamic experiments with a rapid decrease in flow rate suitable for model identification and control design.

The structure of this article is as follows. Section 2 explains the system developed for the experiments, the experimental methods, and the model identification and control procedures. Section 3 shows the results obtained for the three strategies considered: ANN, fuzzy logic, and local models. Section 4 discuss the differences between the control strategies studied. Finally, the Conclusions section summarizes the results and proposes ways forward for further study.

## 2. Materials and Methods

### 2.1. Experimental Setup

Figure 1 shows a schematic of the pilot plant used in the experiments. The feed tank of the main circuit had a volume of 28 L and was tempered in a Polyscience recirculating unit. The main circuit contained a Micro CARBOSEP/Kerasep laboratory module. The peristaltic pump used was the Masterflex I/P^®^ brushless process drive with high-performance pump head able to work from 33 to 650 rpm. The tubing used was Norprene HP 06404-70 with an internal diameter of 9 mm. Downstream of the pump, a pulse damper was placed, constructed from a helical rubber tube. Two Bürket 8323 pressure sensors prepared to work in the 0–4 bar range were placed in the inlet and outlet of the membrane module. A Bürket 8031 flow sensor with a range of 10 to 100 L/h was placed at the membrane outlet to measure the crossflow during fouling experiments and CIP operations. For permeate flow measurement, a device for low flow rates was built ad-hoc. The flow measurement was based on alternate filling and emptying of a permeate collection tank controlled by a solenoid valve and measurement of the position of a buoy with a laser sensor. Two three-way solenoid valves controlled the alternation between a fouling operation step, an alkali cleaning step, and water flushing between both stages. Pump control and acquisition of flow rate and pressure data were performed using a PCB circuit. An Advantech PC1-1711 DAQ card installed in a PC was used to record sensor data and send control actions to the actuators (valves and pump). The user interface was performed in Matlab.

The membrane used in the experiments was the Carbosep M14. This membrane is a tubular ceramic membrane with a length of 400 mm and membrane area of 75.4 cm^2^, a nominal cut-off threshold of 0.14 μm, and a nominal water flow rate of 375 L·h^−1^·m^−2^·bar^−1^.

### 2.2. Experimental Conditions

Model identification experiments were performed using humic acid sodium salt (technical grade from Sigma-Aldrich) at a concentration of 20 mg/L. This concentration was high enough to produce severe fouling in a short time. The experiments were carried out between 2 and 5 h. The temperature of the feed tank was maintained at 25 °C. The applied pressure range to build the model was between 0.3 to 2.5 bar. During the experiments, the inlet and outlet pressure, the concentrate flow rate, and the permeate flow were sampled at a rate of 1 s. The solenoid valve opening percentage and pump rotational speed were the two variables manipulated by the control system during the fouling operating stage to meet the pressure and flow requirements. The control actions were carried out at a rate of 20 s. To recover the permeate flow, a CIP operation was performed after each model identification experiment using a 0.1 mol/L sodium hydroxide solution.

### 2.3. Control Strategies

Three different approaches were used to control the system, requiring different strategies to incorporate the information available from the systems and build the controllers: (i) artificial neural networks, (ii) expert control based on fuzzy logic, and (iii) control by fuzzy-integrated inverse local models.

#### 2.3.1. Control Based on Artificial Neural Networks (ANN)

An artificial neuron is an element that has an internal state (activation level) that receives signals that can change its state. An artificial neural network (ANN) is a structure capable of processing information defined as input variables to obtain a response expressed as output variables. The ANN is organized in layers of neurons in which their parameters (weights *w_i_* and biases *b_i_*) must be determined to match the mapping between inputs and outputs. The fitting procedure to obtain the ANN parameters is known as training. In this case, a non-supervised learning strategy has been used in which the data is divided into one dataset for training, one for validation, and one for testing. Given an experimental output obtained for the input conditions, the errors between the experimental output vector and the response predicted by the ANN in its current state of adjustment against the input are evaluated. The error obtained is used to adjust the values of the weights and biases. The validation data set is used to measure the overall error of the network and determine when to stop training the ANN. In our case, a layer of neurons with linear output and radial activation functions (Figure 2) gave satisfactory results.

The model identified by ANN can be used to predict the behavior of the system, but in this case, the objective was to control the system. Therefore, an ANN-based control technique is needed. The designed controller was based on one of the simplest control ideas that are applicable when a model of the system to be controlled is available: given a desired reference to be followed by a system, the reference passed through the inverse model gives the control action to be injected to the system to achieve the desired reference. In real-world applications, subjected to disturbances, this simple structure is complemented with additional elements. In this case, a classic direct control with reverse pre-feed (see Figure 3) based on two ANN and a reference model is used [31]. The ANN is used twice in this case. In the first step, the error, defined as the difference between the real (Y_comp_) and the desired (ref) output of the system is fed into a predefined reference model (the desired behavior of the plant) giving an output (reff). This output goes to the inverse of the ANN system model, providing then a control action (U_n_) which is compared to the control action (U_PI_) given by a proportional-integral controller (PI) designed for the reference model. This comparison generates the final control action (U) to be fed into the real system. This control action is also fed into the ANN system model, whose output is compared to the real output and the difference feedbacks to the reference. Next, the reference is compensated with the difference between the real output of the plant and the output predicted by a non-affine ANN and this error initiates the control actions computation again.

Therefore, the identification procedure required the identification of two ANNs. First the direct model, which provides the output prediction, was identified. The inputs used to train the ANN were the output in the previous time and the inputs for the plant operation (solenoid valve opening rate and pump rotation speed). Secondly, the inverse model was identified after applying low-pass filtering to the signal. The Neural Network toolbox of Matlab was used for both identifications.

The experimental data were obtained in the working area of the plant for a matrix of combinations of pump speed signals in the range of 0.5–3 V (108–650 rpm) and valve opening in the range of 65% to 80%.

#### 2.3.2. Expert Control Based on Fuzzy Logic

The term fuzzy logic was introduced in 1965 with the Lotfi Zadeh’s proposal of fuzzy set theory [32]. In contrast with Boolean logic which only considers false/truth values of variables expressed by the integers 0 or 1, in fuzzy logic, the level of truth of a variable can be expressed by any real number between 0 and 1. This is a natural way of representing vagueness and imprecise information and has been applied to many fields.

From a practical view, a fuzzy system can be seen as a linguistically interpretable model consisting of several “if-then” rules and logical operators using fuzzy sets as input and output variables:*If input is A then output is B*(1)

A rule-based fuzzy system, although it is a mathematical function and can be identified from data similar to any other system, can make direct use of expert information as long as it is interpretable. Therefore, experts in the operation of a specific system can easily build a model or a controller without any modelling or control knowledge.

In this case, a fuzzy controller was built for the MF system defined above, based on the available knowledge of the experts. The expert (in close collaboration with the control engineer) defined the linguistic variables and the fuzzy rules. The Matlab Fuzzy Systems toolbox was used for implementation purposes.

Although two independent fuzzy controllers can be built to control TMP and retentate flow rate separately, it was thought that a multivariable controller could give better results by taking into account their interaction. Combining expert information and control possibilities, the structure presented in Figure 4 was agreed upon, aiming for fast and accurate control. The main ideas were to quickly drive the system close to the desired TMP/flow operating point, and then fine-tune the control around this point more precisely but more slowly.

The first fast part can be accomplished using a look-up-table (LUT) strategy [33]. The inputs of this table are the desired TMP (P_ref) and the retentate flow rate (Q_r__ref) at the operating point, and its outputs are the valve opening percentage (k_0) and the pump speed (n_0) control actions to reach approximately the reference under standard base conditions.

These base control actions can be modified to new values (*k* and *n*) by a fuzzy controller whose input variables are the transmembrane pressure error and the retentate flow error, thus closing the gap between the actual output results (P and Q_r_) and the desired ones (P_ref and Q_r__ref).

#### 2.3.3. Control by Inverse Local Models

By making use of the expert system information on the operating points defined in the LUT (or any other partition of the input space) defined in the previous section, a more classical control approach can be used. The idea is the opposite of the ANN approach where a very accurate model of the systems is obtained to perform a good control of the plant based on a complex controller. In this case, very simple models of the system must be identified at each of the required operating points and a simple controller must be designed for each of the models. Subsequently, all available control actions are integrated into a single control action depending on the actual operating point. The complete controller can be built, for example, as a fuzzy system [34] in which each controller at each operating point is a rule, and the outputs are interpolated using membership functions created by weighting the distances to the operating points (Figure 5).

Therefore, the idea was to identify several models for relating inputs vs. outputs of the system (i.e., Vel and %Ev vs. P_ref and Q_r__ref in the previous section) that contain only the gain information between each pair of variables at each operating point. The information included in each pair-of-variables local model can be arranged into a matrix where the increments of each output with respect to each input associated with an operating point can be easily organized:(2)$$\left[\begin{array}{c} \Delta P\;\\ \Delta Q \end{array}\right]=\left[\begin{array}{cc} {\frac{\partial P\;}{\partial n\;}|}_{ko} & {\frac{\partial P\;}{\partial k\;}|}_{no}\\ {\frac{\partial Q\;}{\partial n\;}|}_{ko} & {\frac{\partial Q\;}{\partial k\;}|}_{no} \end{array}\right]\cdot \left[\begin{array}{c} \Delta n\\ \Delta k \end{array}\right]$$
where, for the sake of clarity, *n* is pump speed Vel, *k* is solenoid valve opening %Ev, *P* is transmembrane pressure TMP, and *Q* is flow rate Q_r_. In this framework, the derivatives of the local function directly relate the required control actions and the desired change in operating conditions for a particular operating point (*ko, no*).

The controller for both outputs at the same time was easily obtained by inverting the derivative matrix:(3)$$\left[\begin{array}{c} \Delta n\\ \Delta k \end{array}\right]={\left[\begin{array}{cc} {\frac{\partial P\;}{\partial n\;}|}_{ko} & {\frac{\partial P\;}{\partial k\;}|}_{no}\\ {\frac{\partial Q\;}{\partial n\;}|}_{ko} & {\frac{\partial Q\;}{\partial k\;}|}_{no} \end{array}\right]}^{-1}\cdot \left[\begin{array}{c} \Delta P\;\\ \Delta Q \end{array}\right]$$

However, although the simple gain models in each cell of the matrix can easily be inverted (thus having an inverse system model at each operating point for a particular input-output combination), the two-inputs–two-outputs derivative matrix in Equation (2) is not always invertible (values close to zero, or one row/column is linearly dependent on another).

If it is the case and the system is not very fast (as in MF), an approximate multivariable control can be achieved by alternating small steps of single input-output control actions for each of the matrix cells in Equation (3). Figure 6 shows the final control diagram used in the control of the process, where a “control strategy” defines the alternate operation of the single-input–single-output controllers based on the inverse of the appropriate matrix cell.

Recall that there is a controller, as defined in Figure 6, for each predefined operating point *j* in the (*n, k*) plane, and all of them are integrated using the scheme showed in Figure 5. The integration is performed by weighting each control action *j* by its membership function *μ_j_* calculated as the relative distance from the actual operating point of the plant to the local operating point *j* of the model/controller. Whenever our system operates in the (*n, k*) plane of operating points, a particular operating point (*n_i_, k_i_*) of the plant at instant *i* will be surrounded by 4 operating points, and the normalized membership (from 0 to 1) of, e.g., the operating point *i* (out of 4) can be calculated as [35]:(4)$$\mu_{i}=\frac{\frac{1}{d_{i}}}{\sum_{1}^{4}\frac{1}{d_{j}}}$$
where *d* is the absolute value of the Euclidean distance from the actual operating point of the plant to the 4 nearest predefined operating points in the controller.

## 3. Results and Discussion

### 3.1. Control Based on Artificial Neural Networks

The neural network structure used to build the direct and inverse ANN models of the controller had two hidden layers with 6 neurons per layer. The non-affine ANN used as inverse models had an input layer with 8 neurons. This structure permitted a reasonable balance between training time and prediction capability. The best training results were obtained using the Levenberg-Marquardt algorithm. Model identification results were reasonably good for both the direct and inverse models at most operating points.

The controller was tested on a 7 × 7 matrix of equally spaced operating setpoints given by combinations in pump speed signal ranging 0.5–3 V (108–650 rpm) and valve opening percentages from 65% to 80%. Figure 7 shows the trajectories in the flow-pressure space of the controlled pilot plant. The ANN-based control was able to achieve the desired reference (static specifications). However, these satisfactory results could be improved from the dynamic specifications point of view since the controlled system showed significant overshoot and long settling time during the plant start-up at some operating points (Figure 8). This behavior was a consequence of a poor identification of the process dynamics when the network was trained using the entire dataset.

### 3.2. Expert Control Based on Fuzzy Logic

In order to represent the expert knowledge on plant operation in a Mamdani rule-based expert system with a number of fuzzy rules as in Equation (1), the “vocabulary” of fuzzy input and output variables was first defined.

The input variables to the fuzzy controller in Figure 4 were the transmembrane pressure error (eP) and the retentate flow rate error (eQ). First, the ranges of values of the inputs were stablished to obtain the membership functions. The pressure error was divided into the following categories: very negative (VN), negative (N), slightly negative (NS), zero (Z), slightly positive (PS), positive (P), and very positive (VP). The same categories were used to divide the flow rate range. The above definitions, in combination with the information provided by experts, were translated into the fuzzification structure presented in Figure 9.

Linguistic values were also assigned to the controller outputs in Figure 4: pump speed and valve opening. All the linguistic variables are shown in Table 1, and the resulting defuzzification structure is shown in Figure 10.

Once all the linguistic variables and their ranges were defined, the expert in the operation plant was able to translate all his/her knowledge using the limited vocabulary in the expert rules in Table 2. Each row is a possible eQ input, and each column is one of the eP input possibilities. Each cell in the table represents the outputs to be applied. For example, the first cell of the table corresponds to the rule:*If eQ is VN and eP is VN then pump speed is NDH*(5)

Next, a fuzzy rule-based system composed of 49 rules as described above was then defined to perform the control of the MF plant.

The combination of all the information provided by the expert (linguistic variables and rules) resulted in the nonlinear response surface of the controller for both outputs. The controller performance can be seen for different operating points in Figure 11, with lower overshoot and settling time than in the ANN approach for the same operating conditions.

The performance of the controller was also good for long experiments (about 4 h), where the effect of fouling changed the plant behavior, as shown in Figure 12. The experiments were repeated several times. The figure shows 3 cycles for pressure and flow control at the operation point (55 L/h, 1.5 bar).

Finally, some experiments were performed to see the behavior of the controller when changing the operating point. The results are shown in Figure 13.

### 3.3. Control by Fuzzy-Integrated Inverse Local Models

The 49 operating points, defined in the previous section, covering the entire operating space of the plant were also considered in this study. A local model was identified for each of them following Equation (2), and a local controller designed as in Equation (3).

In order to reach a desired operating point (P_ref, Q_r__ref) while avoiding the inversion problem in Equation (3), the alternate operation of the two controllers shown as “control strategy” in Figure 6 was:Reach Q_r__ref using the pump speed control variable Vel, but with the solenoid valve fully open (for security reasons)If more pressure is needed, then close the solenoid valve by controlling the valve opening %Ev to reach P_refThe last action will decrease the flow. Then reach again Q_r__ref using the pump speed control variable Vel.Follow the last two steps until both references (P_ref, Q_r__ref) are obtained.

The desired operating point (P_ref, Qr_ref) need not be exactly one of the 49 predefined operating points. Therefore, an interpolation strategy of the control actions was defined (Equation (4)). Figure 14 shows, in the flow-pressure plane, how a representative operating point (P_ref = 2 bar, Q_r__ref = 54 L/h) was reached using the proposed strategy.

Figure 15, Figure 16, Figure 17 and Figure 18 then show the evolution over time of the outputs and control actions in the same experiment. The results of the new strategy (inverse local models) are plotted in blue, while the best strategy so far (expert control based on fuzzy logic) is plotted in red for ease of comparison.

### 3.4. Discussion of the Results

The three chosen approaches (ANNs, expert control, and fuzzy-integrated local inverse models) were able to adequately control the MF system, as shown in the previous results subsections. However, the three controllers showed differences in the performance of the dynamic specifications: the speed at which the system reaches the operating point (settling time), and presence of oscillations (overshoot). Some differences were also found in terms of their robustness, and the data required for the development of the controllers (Table 3).

The ANN-based control had some problems at different operating points showing oscillations that can be attributed to incorrect identification of the dynamics when the network is trained to meet the full range of operating conditions. However, in this case much more data is needed, or a more complex network structure may be required. This implies a higher training effort which may make it difficult to use an adaptive approach.

The control based on fuzzy rules was able to work quickly from the fuzzy rules deduced from the plant operating experience. However, it has the problem that in case of major changes in the system, the expert will be needed again to define the new rules and the whole identification process will have to be repeated.

Local models have the great advantage of requiring less data and expert effort than the other methods for their identification. The settling time is somewhat slower than that achieved by the expert model but sufficient for the studied process. Moreover, compared to the expert models, they allow an easy adaptive implementation of the control, being then the best simplicity-performance trade-off of the AI approaches shown.

## 4. Conclusions

All three types of regulators implemented were functional and capable of accurately achieving the static specification reference over a wide range of flow and pressure operating conditions. The controllers were also able to function properly during the period when the system was subjected to a high degree of fouling whenever the membrane of the system was cleaned after each operation. The controllers performed well during experiments in which the cleaning operations were able to avoid high irreversible fouling. Therefore, it can be said that the AI control approaches studied in this work showed very good performance in the control of a MF process.

The proposed control techniques can be applied to different fields: separation in the food industry, membrane bioreactors, membrane medical applications, and laboratory-scale experimentation. The technique to be used in each specific case depends mainly on the information available: ANNs require a lot of experimental data, expert systems require a human expert to be available, and local models require a combination of some expert information and a few data. Dynamic process control requirements may also decide the most appropriate approach, as some techniques are faster than others, in this case expert control being the fastest technique. The same is true for robustness with expert systems providing the best results in this case as well.

Although the proposed control strategies worked well under the conditions of the in the experiments, adaptive versions of the three approaches are desirable and will be developed in the future. Adaptive strategies will allow on-line calibration of the model and controller during operation in the event of irreversible fouling or other changes in the system, such as changes in temperature, pressure, or feed composition. In most situations, the effects on process performance of operating pressure, crossflow velocity, temperature, and pH are complex, highly feed composition dependent, and sometimes weakly explained by physical models. A desirable adaptive control would be able to adjust in minimum time the operating variables seeking to optimize process performance. For example, such controllers can be used in a batch process to adjust pressure and crossflow velocity to minimize the energy required for the concentration stage. Alternatively, they can also be used in continuous processes to jointly optimize the operation phase and the cleaning phase to improve process economics.

## Figures and Tables

**Figure 1 membranes-12-00883-f001:**
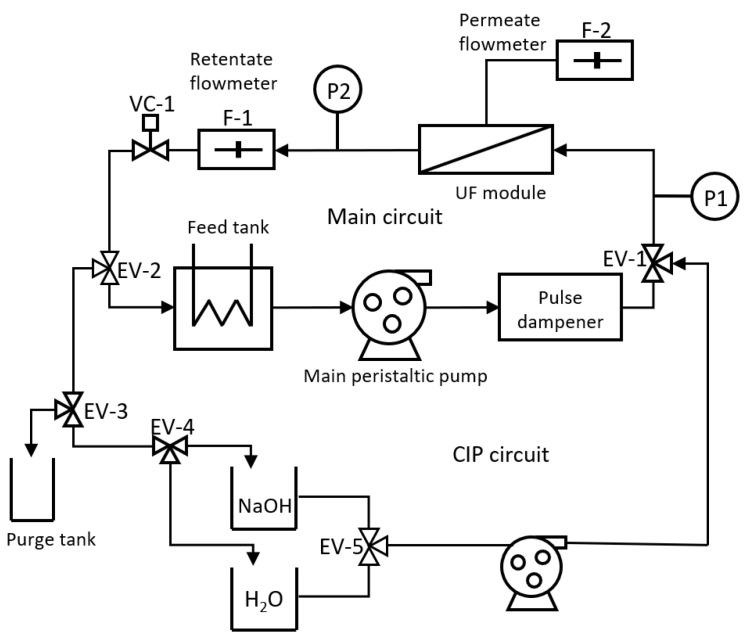
Schematic of the pilot plant.

**Figure 2 membranes-12-00883-f002:**
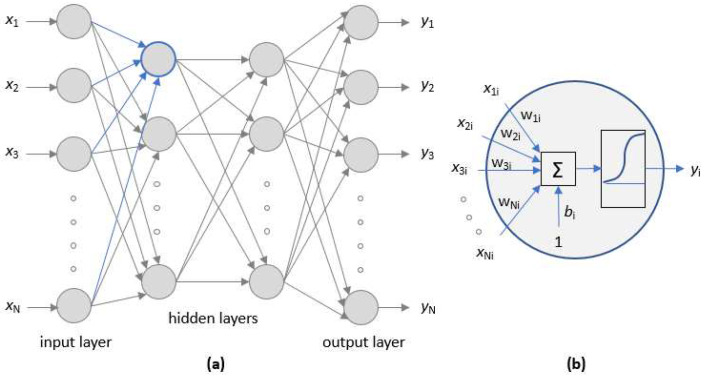
Artificial neural network: (**a**) layers structure, (**b**) detail of a neuron.

**Figure 3 membranes-12-00883-f003:**
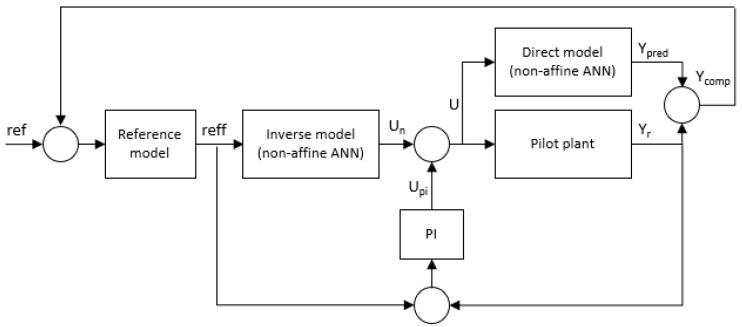
Direct control with reverse pre-feed based on artificial neural networks.

**Figure 4 membranes-12-00883-f004:**
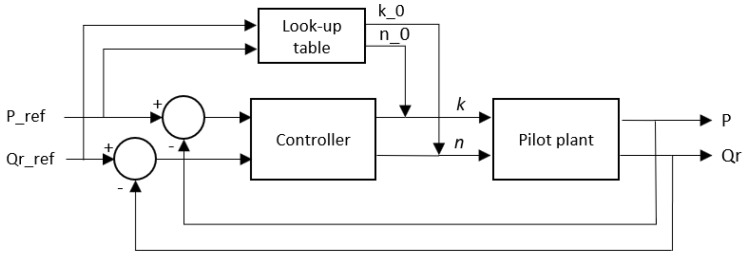
Diagram of the fuzzy controller.

**Figure 5 membranes-12-00883-f005:**
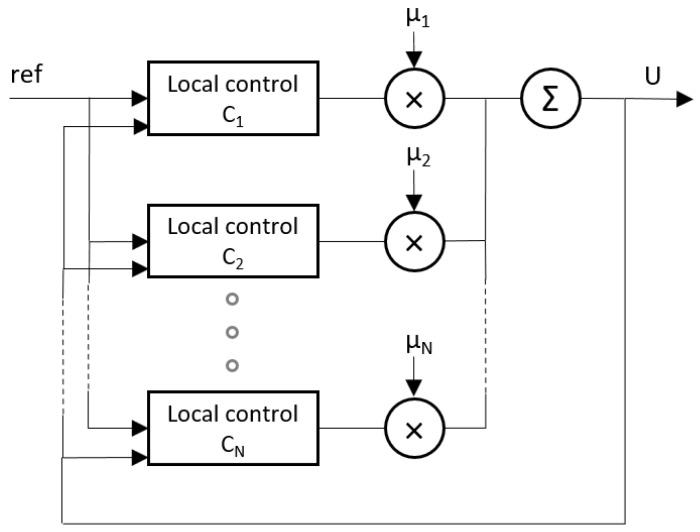
Control structure with local models.

**Figure 6 membranes-12-00883-f006:**
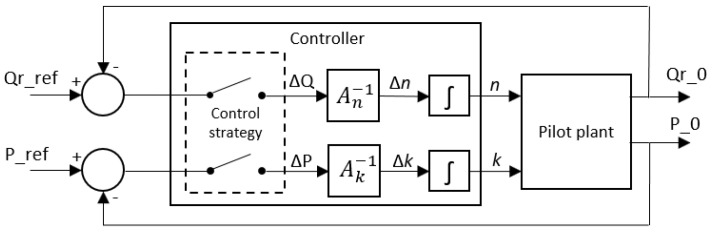
Global control diagram in an operating point.

**Figure 7 membranes-12-00883-f007:**
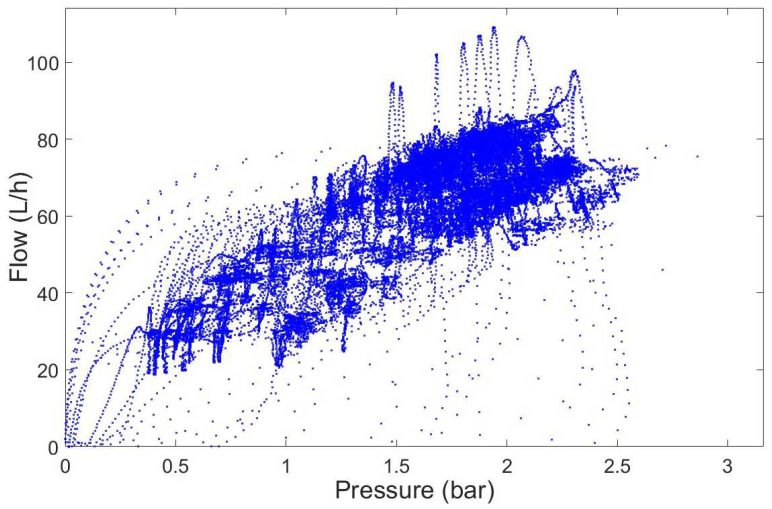
Operating working area of the pilot plant obtained for pump signal speed 0.5–3 V (108–650 rpm) and valve opening 65–80%.

**Figure 8 membranes-12-00883-f008:**
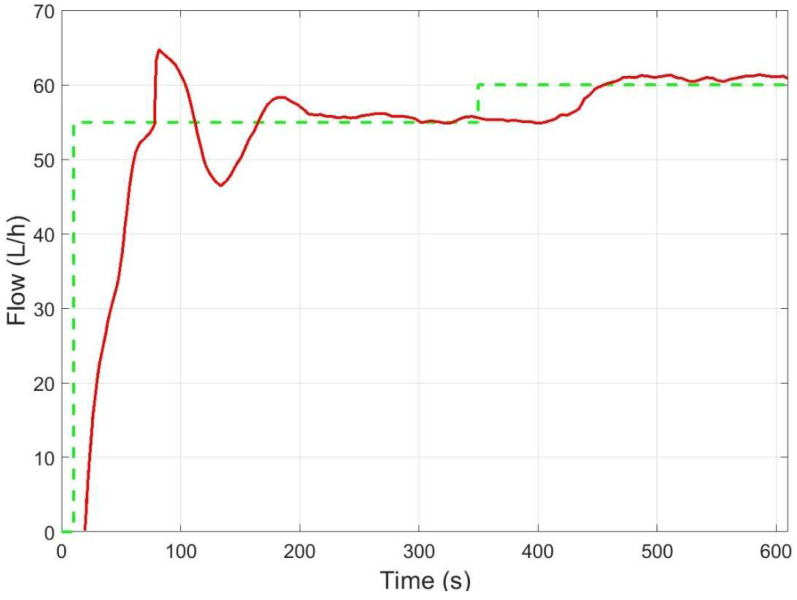
Control dynamics of ANN controller (flow rate reference in dashed green and flow output in red).

**Figure 9 membranes-12-00883-f009:**
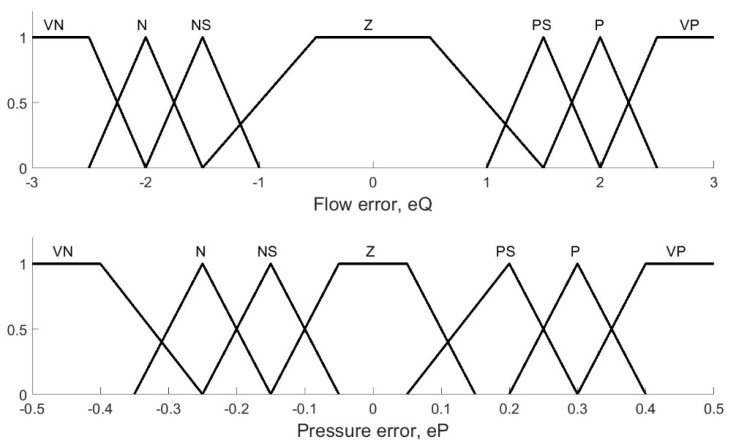
Linguistic values assigned to the controller inputs.

**Figure 10 membranes-12-00883-f010:**
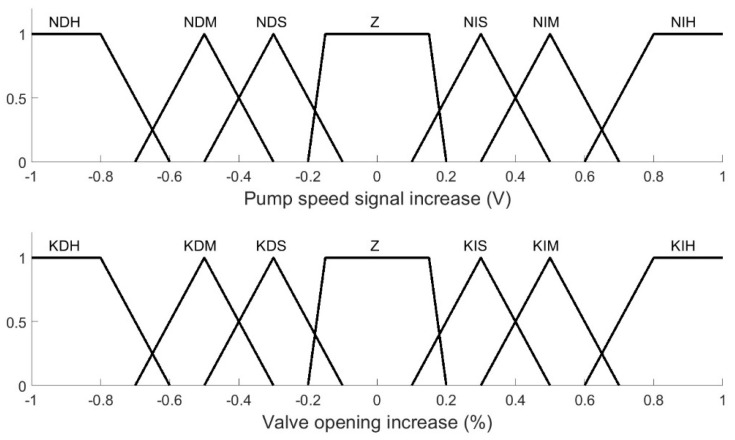
Linguistic values assigned to the controller outputs.

**Figure 11 membranes-12-00883-f011:**
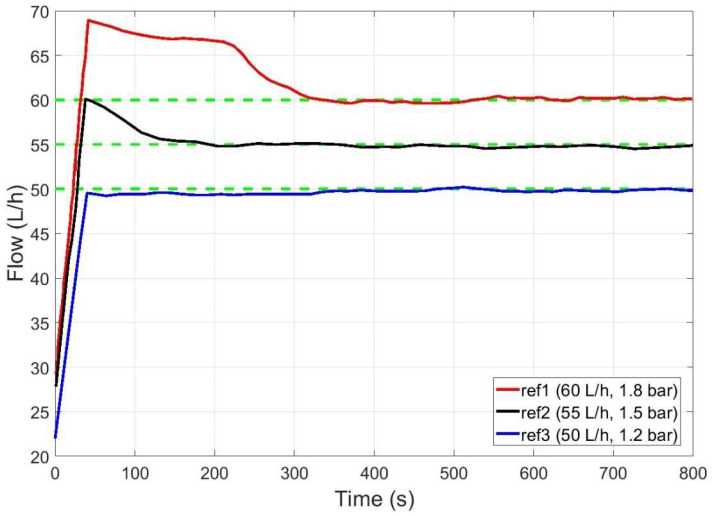
Controller performance for flow rate at different operating points defined by flow rate and pressure (flow references are marked in dashed green in all cases).

**Figure 12 membranes-12-00883-f012:**
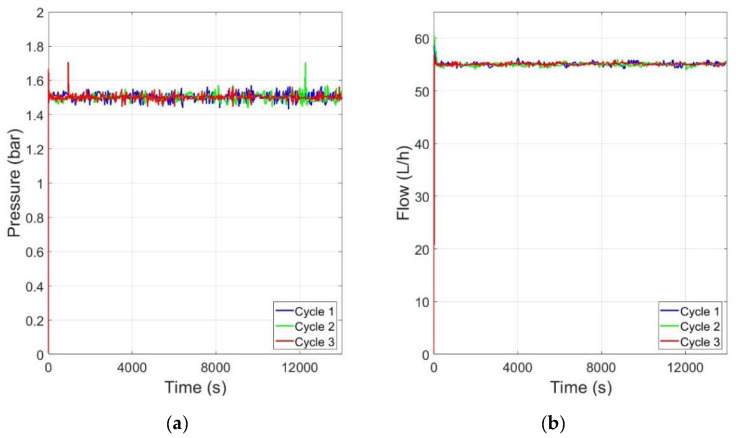
Controller performance in long time runs at the operation point (55 L/h, 1.5 bar): (**a**) operating point for pressure; (**b**) operating point for flow rate.

**Figure 13 membranes-12-00883-f013:**
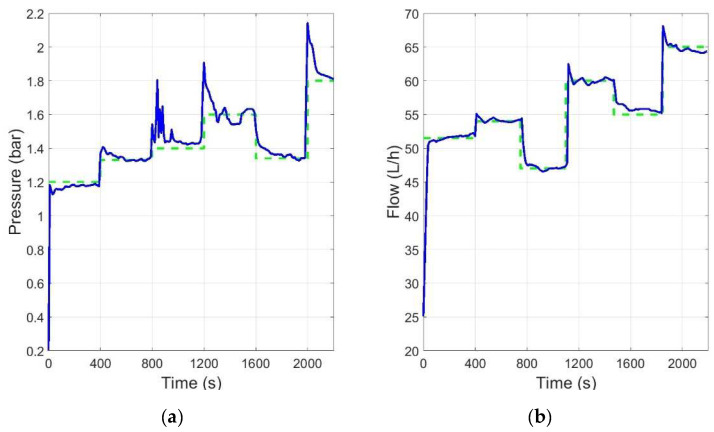
Controller performance (reference in green and output in blue): (**a**) change of the operating point for pressure; (**b**) change of operating point for flow.

**Figure 14 membranes-12-00883-f014:**
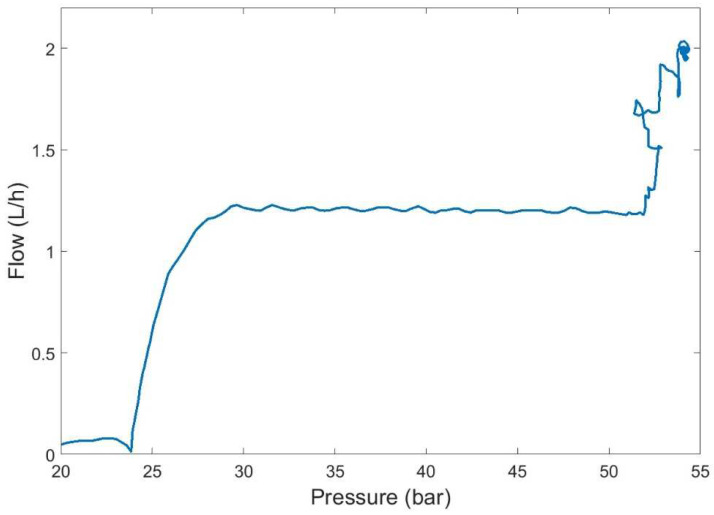
Flow-pressure path followed to reach a predefined operating point (P_ref = 2 bar, Q_r__ref = 54 L/h) using the proposed strategy.

**Figure 15 membranes-12-00883-f015:**
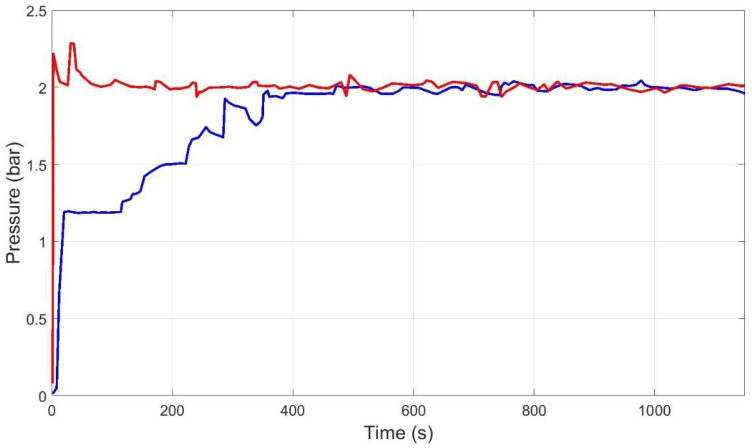
Evolution of pressure output to reach the representative operating point (P_ref = 2 bar, Q_r__ref = 54 L/h) using inverse local models (blue), and expert control based on fuzzy logic (red).

**Figure 16 membranes-12-00883-f016:**
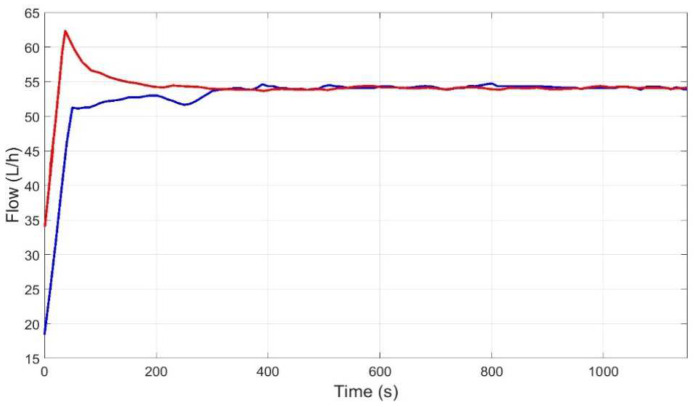
Evolution of flow output to reach the representative operating point (P_ref = 2 bar, Q_r__ref = 54 L/h) using inverse local models (blue), and expert control based on fuzzy logic (red).

**Figure 17 membranes-12-00883-f017:**
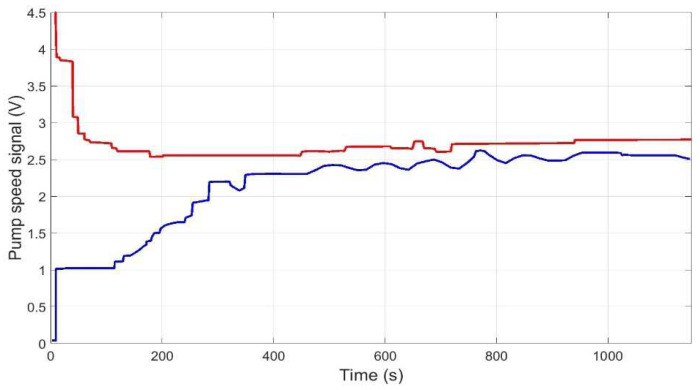
Evolution of pump speed control action to reach the representative operating point (P_ref = 2 bar, Q_r__ref = 54 L/h) using inverse local models (blue), and expert control based on fuzzy logic (red).

**Figure 18 membranes-12-00883-f018:**
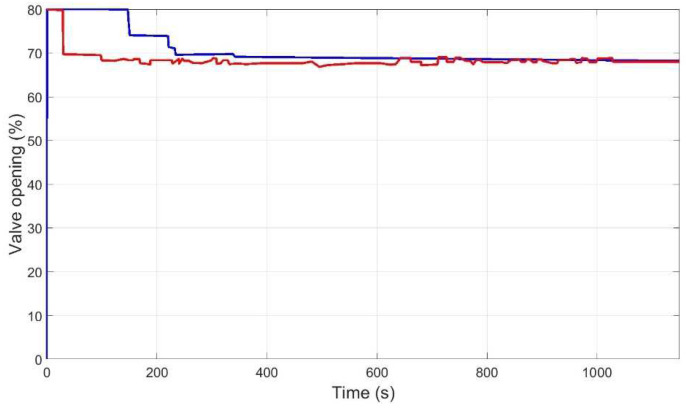
Evolution of valve opening control action to reach the representative operating point (P_ref = 2 bar, Q_r__ref = 54 L/h) using inverse local models (blue), and expert control based on fuzzy logic (red).

**Table 1 membranes-12-00883-t001:** Linguistic variables of the controller output.

	Pump Speed	Valve Opening
increase (high)	NIH	KIH
increase (medium)	NIM	KIM
increase (slight)	NIS	KIS
decrease (slight)	NDS	KDS
decrease (medium)	NDM	KDM
decrease (high)	NDH	KDH

**Table 2 membranes-12-00883-t002:** Expert rules for the fuzzy controller according to defined fuzzy variables for inputs and outputs.

eQ/eP	VN	N	NS	Z	PS	P	VP
VN	NDH	KDM NDM	NDM	KDM NDS	NIS	KDM	KDM NIM
N	NDH	NDM	NDM	KDM	KDS NIS	KDM	KDM NIS
NS	NDM	NDM	NDS	KDS	KIS	KIM	NIM
Z	KIS	NDM	KIS	-	KDS	NIM	NIM KDS
PS	KIM	KIM	KIS	KIS	KDS NIS	NIM	NIM
P	KIH	KIM	KIM	KIM	NIM	NIM	NIM
VP	KIH	KIS NDS	KIS	KIS NIS	KDS	KDS	NIH

**Table 3 membranes-12-00883-t003:** Comparison of the control techniques studied.

Controller	Modelling	Control	Robustness
Local models	Few data required	Slow, precise	High. Some problems in different functioning points
Expert control	Only models look-up-table	Sharp, fast, precise	Pondering problems in different working areas
Artificial neural networks	Many data required	Slow, precision achieved by PI controller	Limited. The models do not cover all areas

## Data Availability

Not applicable.

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
