# Peer review of "Comparison of Artificial Intelligence Control Strategies for a Peristaltically Pumped Low-Pressure Driven Membrane Process"

_membranes, 2022, doi:10.3390/membranes12090883_

Round 1

Reviewer 1 Report

This article could be published in Membrane, because the three types of controllers that were implemented were functional and able to achieve the reference with precision (static specifications) in a wide range of operating conditions (crossflow and pressure) in MF process.

1. Figure 11:  "black (50 L/h, 1.2 bar)" seems to be  blue (50 L/h, 1.2 bar).

Reviewer 2 Report

Reviewer comments_Membranes-1881494

The authors of this manuscript reported on the comparison of artificial intelligence control strategies for a peristaltically pumped microfiltration process. The findings reported in this manuscript deserve to be known by other researchers. However, this manuscript requires major revision, and before publication, several questions should be illustrated more clearly to make the manuscript more readable and meaningful to readers. Detailed comments are as follows:

  1. Kindly check all the grammar/tenses used in this manuscript. Please use the past tense whenever required.
  2. Kindly provide recommendations for this study, its future way forward and the contribution of this study to the body of knowledge.
  3. Restructure the manuscript into Introduction, Materials and Methods, Result and Discussion and lastly, Conclusions. The content discussed should tally with the section’s topic. So that easy for readers to understand the storyline and appreciate the contents.
  4. Shorten the lengthy sentence and revise them. For example: ‘The best results, according to the required process dynamics, among the approaches studied were obtained with the use of local models as this method.
  5. Revise the abstract according to the research background, problem statement, method, result, discussion and conclusion so that readers can grab the helicopter view of this study.
  6. Graphs in Figures 8 and 11, 15, 16, 17, and 18 are blurry. Improve the resolution
  7. No conclusion section is provided.
  8. The title contradicts the content of the manuscript. For example, is the experiment run as UF or MF? Kindly revise the title of the manuscript.
  9. The language in this manuscript needs to be improved, considering the frequent types and confusion caused by inaccurate phrases and sentences.
